# Role of Situational Dependence in the Use of Self-Service Technology

**Cheolho Yoon [1],\* and Byongcheon Choi [2]** 

[1]  Department of Business Administration, Mokpo National University, 61 Dorim-ri, Chungkye-myun, Muan-gun, Jeonnam 58554, Korea
[2]  Department of Medical Information, Iksan General Hospital, Jeollabuk-do 54543, Korea ; gaztomy@hanmail.net
\*  Correspondence: carlyoon@empal.com

**Abstract:** Although numerous studies have been conducted on the use of self-service technology (SST), little is known about the importance of the situation for individuals' acceptance of SSTs. This study proposed the situational dependency construct and analyzed the role of the construct in the use of SSTs. To conduct an empirical analysis, a research model combining the situational dependency variable with the perceived usefulness and perceived ease of use of the technology acceptance model variables, which are the most frequently used in studies related to SSTs, was developed. A total of 213 valid data were collected through questionnaires and analyzed using confirmatory factor analysis and path analysis through structural equation modeling. The results showed that situational dependence influenced attitude toward using along with perceived ease of use and perceived usefulness and that situational dependence had a strong influence on intention to use. This study provides strategic insight for practitioners to lead acceptance of SSTs.

**Keywords:** self-service technology; situational dependence; technology acceptance model; situational factors

## 1. Introduction

With the rapid development of information and communication technologies and the need to reduce the cost of providing services, companies are rapidly introducing and operating self-service technologies (SSTs). SSTs are technological interfaces that provide services to consumers themselves by using the technology without direct contact with service employees [1]. SSTs enable customers to obtain fast service processing by reducing wait times [2] and enabling companies to benefit from reduced costs [3]. Despite these obvious benefits, there are people who still hesitate or avoid using SSTs [4]. A number of studies have been conducted to explain or promote the use of SSTs in various fields, such as information systems, consumer behavior, and marketing.

Most of the research related to the acceptance of SSTs so far focuses on the characteristics of the technology or the traits of the individual who uses it. Moreover, only a few studies have been conducted on the situational influence at that point in time using SST. Research on situational influences in the traditional services literature is continuing, but because of the significant differences in the processes between traditional services by employees and self-service based on technologies, examining the situational influences of SSTs is necessary [5]. Therefore, this study develops a situational dependence construct based on situational factors to analyze the situational influence in using SSTs. Moreover, it empirically analyzes the effect of situational dependence on perceived usefulness and perceived ease of use through the technology acceptance model (TAM), which is frequently used in the SST research, in regards to the attitude and usage of SSTs.

The results of this study are anticipated to be utilized as a theoretical foundation for constructing a research model in subsequent studies to analyze various situational factors. They are also expected to provide implications for the activation of SSTs by emphasizing the importance of the situation in practice.

## 2. Theoretical Background

### 2.1. Self-Service Technology

SST refers to "technological interfaces that enable customers to produce a service independent of direct service employee involvement" [1] (p. 50). Traditional service encounters are interactions between customer and service employees through face-to-face contact. However, technology-based self-service with interaction between customer and technologies is gradually increasing because of the advances in technologies.

SSTs provide positive effects for both businesses and consumers. Companies can reduce their financial costs by reducing the number of service employees and increase consumer satisfaction by employing a standardized service delivery and matching customer expectations with delivery service [1]. Consumers obtain benefits from the use of SSTs, such as saved time and money, increased accessibility, a variety of choices, convenience, reduced wait times, high levels of personalization, and fast service processing [2]. However, as these positive effects can occur when consumers use SSTs directly, it is more important for customers to use SSTs. Therefore, most studies on SST have been focused on consumers' acceptance of SSTs [6–8]. Table 1 shows the factors affecting the acceptance of SSTs presented in existing studies. As shown in Table 1, most existing studies on the acceptance of SSTs have focused on the characteristics of the technology (e.g., perceived usefulness and perceived ease of use) [6], consumer traits (e.g., technology anxiety) [9], and self-efficacy [10] as the factors affecting an individual's acceptance of SSTs. Only a few researchers [5,11,12] have examined the situational influence of using customers' SSTs. Dabholkar and Bagozzi (2002) proposed perceived waiting time and social anxiety as situational factors and analyzed their moderating effects [11]. Gelderman et al. (2011) proposed the role of clarity and perceived crowdedness as situational factors and analyzed their effects on the usage of SST [12]. The study found that both situational variables affected the usage of SST. Collier et al. (2015) analyzed the presence of employees, order size, location convenience, and tolerance to wait as the factors affecting perceived time pressure, which is a situational factor in accepting SST [5]. A few of these studies suggested that situational factors could be important factors in the use of SSTs in addition to the characteristics of the technology and customer traits.

**Table 1.** Factors affecting an individual's acceptance of self-service technologies (SSTs).

| Classification | Factor | Researcher |
|---|---|---|
| Characteristics of Technology | perceived usefulness/performance | [6,8,11,13–16] |
| | perceived ease of use | [6,8,11,13–16] |
| | compatibility | [16] |
| | reliability | [7,15–18] |
| | newness | [15] |
| | comfort | [18] |
| | delivery speed | [18] |

**Table 1.** *Cont.*

| Classification | Factor | Researcher |
|---|---|---|
| Consumer Traits | technology anxiety | [8,9,19] |
| | technology readiness | [7,12,13] |
| | enjoyment/fun | [7,11,13,15,16,18] |
| | self-efficacy | [10,14,16] |
| | subjective norms | [16] |
| | perceived control | [16,18] |
| | the need for interaction | [6,12,17,20,21] |
| | sensation-seeking | [22] |
| | innovativeness | [22–24] |
| | self-consciousness | [11] |
| Situational Factors | perceived waiting time | [11] |
| | perceived crowdedness | [12] |
| | role clarity | [12] |
| | social anxiety | [11] |
| | perceived time pressure | [5] |

## 2.2. Situational Dependence

Dependence is the degree to which the other party needs something to achieve its purpose [25]. Therefore, in this study, situational dependence refers to the extent to which a person is perceived to rely on a particular technology because of its situational factors. That is, situational dependence is based on the effects of situational factors. In the marketing literature, situational factors refer to "all those factors particular to a time and place of observation which do not follow from a knowledge of personal and stimulus attributes and which have a demonstrable and systematic effect on current behavior" [26] (p. 152). Belk (1975) proposed the concept through the definition of existing research and revealed its five dimensions: physical surroundings, social surroundings, temporal perspective, task definition, and antecedent states. Researchers have argued that situational factors such as time constraints, complexity of place, and task definition can change people's behavior; for example, shortage of time may reduce or stop both planned and unplanned purchases [27], purchase behavior can be changed by the location of the store [28] and product display [29], and gifts (task definition) can make people buy expensive things they do not usually purchase.

Situational factors not only change people's behavior but also cause people to perform certain actions. For example, using the ATM in a bank where many people are waiting to receive services, processing with a computer system to complete tasks that must be handled through multiple windows (e.g., visa issuance), and ordering at a kiosk at a fast food restaurant where no one is taking orders can be considered as instances of people using SSTs because of situational factors. That is, people seem to rely on SSTs because of such situational factors as time perspective, method perspective, and task-processing perspective. Therefore, we defined the three dimensions of situational dependency as follows:

- Time dimension: the degree to which people rely on SSTs because of the perception that it would take a long time to process a task without using SSTs in the situation;
- Method dimension: the degree of dependence on SSTs due to the perception that the methods and procedures for performing a task are complicated and difficult without using SSTs in the situation;
- Task-processing dimension: the degree of dependence on SSTs due to the perception that it is impossible to achieve a task without using SSTs in the situation.

## 3. Research Model and Hypotheses

### 3.1. Research Model

The TAM proposed by Davis (1989) is the most commonly used basic theory in the acceptance research of SSTs [30]. Today, TAM theory uses a mixture of the original version, which includes the attitude variable, and a version excluding the attitude variable [31]. Although recent studies using the TAM mainly use the version excluding the attitude variable, in the case of SSTs, many studies ([6,11,14,20,32]) used the attitude variable because of the technical characteristics of SSTs and the high explanatory power of attitude toward intention to use. Therefore, this study also uses the attitude variable. As shown in Table 1, perceived usefulness and perceived ease of use, two key variables of the TAM, were jointly presented in almost all studies. Therefore, we proposed a research model that combines situational dependency with perceived usefulness and perceived ease of use of the TAM variables. Figure 1 represents this research model.

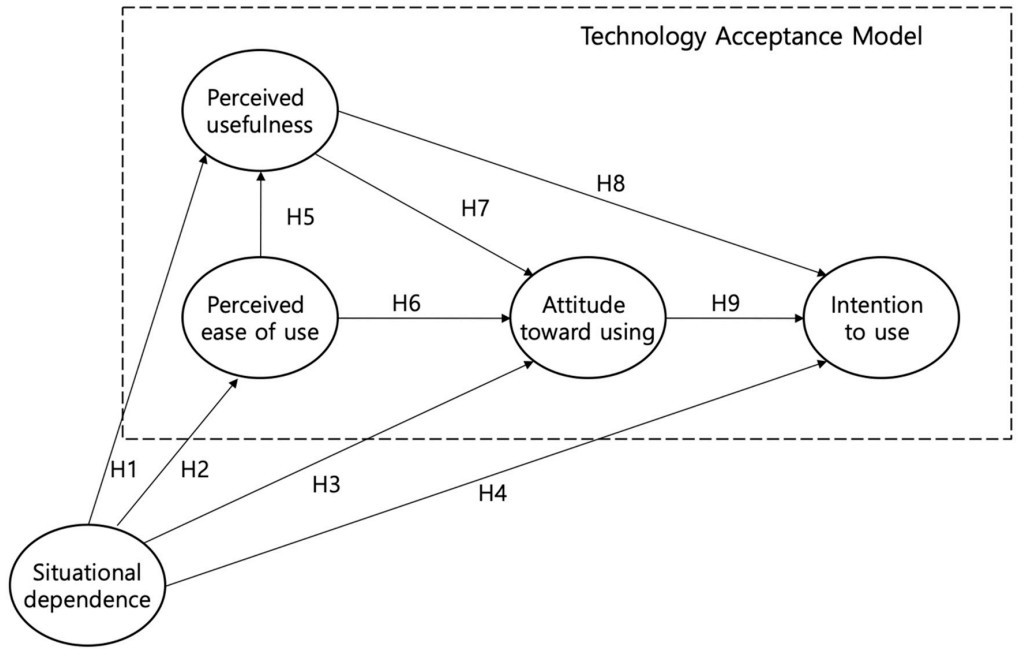

**Figure 1.** Research model.

### 3.2. Hypotheses

Situational dependence on SST can be defined as the extent to which people perceive it as desirable to rely on SST because of situational factors, such as time, usage methods, and task processing. In other words, SSTs can be a useful alternative for handling tasks because of situational factors. For example, using a kiosk that is an empty SST at waiting order reception windows is considered a useful way to reduce an individual's ordering time, and a vending machine that sells cigarettes at the times when the convenience store is closed is useful to people. Moreover, a patient can easily book a medical treatment by touching the booking button with a large image on the touchscreen using a kiosk. People have a favorable attitude toward the perceived usefulness of SSTs. That is, situational dependence on SST is expected to have a positive effect on perceived usefulness, perceived ease of use, and attitude to use it. Therefore, we present the following hypotheses:

**Hypothesis 1 (H1).** *Situational dependence positively affects perceived usefulness.*

**Hypothesis 2 (H2).** *Situational dependence positively affects perceived ease of use.*

**Hypothesis 3 (H3).** *Situational dependence positively affects attitude toward using.*

In real life, situational dependence shows that it directly affects the use or intention to make use of SSTs. When six check-out lines are reduced to two lines in a superstore, people have been observed moving to the self-service checkout windows. When check-in lines are long at the airport, people flock to use the automatic check-in devices. Therefore, the situational dependence on SST is expected to directly affect intention to use. Therefore, we establish the following hypothesis:

**Hypothesis 4 (H4).** *Situational dependence positively affects intention to use.*

The TAM [33], which focuses on identifying the variables affecting users' use and behavior in a technology-related product or service environment, is a widely used theory in SST acceptance studies [30]. An important concept in the TAM is users' belief in the technology, and perceived usefulness and perceived ease of use belief variables have been found to significantly affect consumer attitude and intention to use SSTs in previous SST acceptance studies (Table 1). Thus, in this study, the following hypotheses were established on the basis of the existing TAM theory:

**Hypothesis 5 (H5).** *Perceived ease of use positively affects perceived usefulness.*

**Hypothesis 6 (H6).** *Perceived ease of use positively affects attitude toward using.*

**Hypothesis 7 (H7).** *Perceived usefulness positively affects attitude toward using.*

**Hypothesis 8 (H8).** *Perceived usefulness positively affects intention to use.*

**Hypothesis 9 (H9).** *Attitude toward using positively affects intention to use.*

## 4. Research Methodology

### 4.1. Data Collection

Data were collected from self-service kiosk users in hospitals in Korea. For the online survey, papers with questionnaire guides and mobile-linked QR codes were distributed to self-service kiosk users in the hospitals. Through the questionnaire, 213 questionnaires were collected and used for an empirical analysis of the research model. According to the gender response to the survey, 133 men and 80 women were surveyed, and 82% of the respondents were over 40 years old and 35% of the respondents were company workers. Detailed demographic statistics for the respondents are shown in Table 2.

**Table 2.** Descriptive statistics of respondents' characteristics.

| Measure | Value | Frequency | Percentage |
|---------|-------|-----------|------------|
| Gender | Male | 133 | 62.4 |
| | Female | 80 | 37.6 |
| | - | 213 | 100.0 |
| Age | 20–24 | 2 | 0.9 |
| | 25–29 | 9 | 4.2 |
| | 30–34 | 12 | 5.6 |
| | 35–39 | 16 | 7.5 |
| | 40–44 | 46 | 21.6 |
| | 45–49 | 52 | 24.4 |
| | Older than 50 | 76 | 35.7 |
| | - | 213 | 100.0 |

**Table 2.** *Cont*.

| Measure | Value | Frequency | Percentage |
|---|---|---|---|
| Job | None | 2 | 0.9 |
| | Student | 1 | 0.5 |
| | Company worker | 74 | 34.7 |
| | Housewife | 26 | 12.2 |
| | Specialized job | 62 | 29.1 |
| | Other | 48 | 22.5 |
| | - | 213 | 100.0 |

*4.2. Measurement Development*

The measurements for testing the research model were developed based on the items which were verified as reliable and valid in previous studies. The measurements for perceived ease of use and perceived usefulness were adapted from Davis' studies (1989), which established their reliability and validity. The items for situational dependence were newly developed in this study and were based on the three dimensions of situational dependency, namely, time, method, and task processing, as defined earlier in the theoretical background section. All items were measured using a seven-point Likert scale, with responses ranging from "strongly disagree" to "strongly agree". The measurements used in this study are shown in Appendix A.

*4.3. Data Analysis Techniques*

This study uses structural equation modeling (SEM) for analysis of the research model. The SEM, a statistical analysis method mainly used for causality analysis today, is divided into covariance-based structural equation modeling (CB-SEM) and partial least squares structural equation modeling (PLS-SEM). Unlike CB-SEM, which is mainly used for confirmatory research, PLS-SEM is advantageous for conducting exploratory studies and analyzing complex models because the requirements for sample size or residual distribution are relatively less stringent [34]. As a result, PLS-SEM is actively used in various fields these days. Since this study has an exploratory nature and PLS-SEM is more flexible than CB-SEM in conducting a study, this study employs the PLS-SEM approach. To utilize the PLS-SEM technique, the plspm-package of open-source software R developed by Sanchez (2013) was used [35].

## 5. Results

*5.1. Reliability and Validity Assessments*

This study used Cronbach's alpha coefficient and composite construct reliability (CCR) values for the reliability evaluation of measurement items. Table 3, a reliability assessment, shows that Cronbach's alpha coefficients for all constructs are greater than 0.70 and that the values of CCR are also well above the 0.70 level recommended by Bagozzi & Yi (1988) [36]. Therefore, the reliability assessment results indicated that there is confidence in all measurement items.

**Table 3.** Reliability.

| Construct | Item No. | C. Alpha * | CCR ** | AVE *** |
|---|---|---|---|---|
| Perceived Ease of Use | 3 | 0.882 | 0.928 | 0.810 |
| Situational Dependence | 3 | 0.799 | 0.882 | 0.714 |
| Perceived Usefulness | 3 | 0.935 | 0.958 | 0.884 |
| Attitude toward Using | 2 | 0.863 | 0.936 | 0.879 |
| Intention to Use | 2 | 0.902 | 0.953 | 0.911 |

* Cronbach's alpha, ** Composite Construct Reliability, *** Average Variance Extracted.

This study conducted the convergent and discriminant validity tests using a confirmatory factor analysis (CFA) to assess the validity of the constructs. In the convergent validity evaluation using a CFA, when the *t*-values of the measurement items are significant, that is, when the *p*-value is greater than 0.05, it is evaluated as convergent validity [37]. It is also evaluated that there is validity when average variance extracted (AVE) of the constructs exceeds 0.50 [38]. Table 4, which gives the results of confirmatory factor analysis, shows that all the *t*-values of the measurements are above 1.96, and the AVE values of all constructs in Table 3 are over 0.7. Therefore, the convergent validity assessment results indicated that there is validity in all measurement items.

**Table 4.** Results of confirmatory factor analysis.

| Items | PE | SD | PU | AT | IU | *t*-Value |
|---|---|---|---|---|---|---|
| a4 | 0.94 | 0.61 | 0.66 | 0.63 | 0.61 | 89.44 |
| a5 | 0.93 | 0.62 | 0.63 | 0.65 | 0.65 | 67.81 |
| a6 | 0.84 | 0.62 | 0.57 | 0.48 | 0.50 | 28.17 |
| a7 | 0.52 | 0.84 | 0.54 | 0.64 | 0.65 | 31.02 |
| a8 | 0.66 | 0.86 | 0.55 | 0.62 | 0.63 | 39.67 |
| a9 | 0.56 | 0.83 | 0.54 | 0.54 | 0.60 | 26.5 |
| a1 | 0.64 | 0.57 | 0.94 | 0.67 | 0.61 | 71.37 |
| a2 | 0.63 | 0.58 | 0.94 | 0.67 | 0.64 | 81.27 |
| a3 | 0.68 | 0.65 | 0.95 | 0.70 | 0.69 | 84.3 |
| a10 | 0.59 | 0.63 | 0.66 | 0.93 | 0.76 | 53.22 |
| a11 | 0.64 | 0.71 | 0.70 | 0.95 | 0.89 | 108.78 |
| a12 | 0.64 | 0.73 | 0.69 | 0.88 | 0.96 | 154.52 |
| a13 | 0.61 | 0.69 | 0.62 | 0.81 | 0.95 | 103.91 |

PE: Perceived Ease of Use, SD: Situational Dependence, PU: Perceived Usefulness, AT: Attitude toward Using, IU: Intention to Use.

In the discriminant validity evaluation using a CFA, validity is ensured when the measurement items are loaded more strongly on the allocated construct than on other constructs in the CFA and when the correlation value between the constructs is less than the square root of AVE of their constructs [37]. Table 4 shows that all measurement items are loaded higher in their constructs than in other constructs. Table 5 shows that the square root values of AVE of all constructs are larger than correlations between their constructs. Therefore, the discriminant validity assessment results showed that there is validity in all constructs.

**Table 5.** Square root of average variance extracted (AVE) and the correlation matrix.

| Construct | PE | SD | PU | AT | IU |
|---|---|---|---|---|---|
| Perceived Ease of Use | (0.9) | | | | |
| Situational Dependence | 0.69 | (0.85) | | | |
| Perceived Usefulness | 0.69 | 0.64 | (0.94) | | |
| Attitude toward Using | 0.66 | 0.71 | 0.73 | (0.94) | |
| Intention to Use | 0.65 | 0.74 | 0.69 | 0.89 | (0.95) |
| Mean | 5.22 | 4.62 | 4.47 | 4.99 | 4.79 |
| Standard Deviation | 1.39 | 1.38 | 1.38 | 1.44 | 1.57 |

( ): Square root of AVE.

## 5.2. Hypothesis Testing

For hypothesis testing, we set up a research model as a structural model and performed structural equation modeling. Figure 2 shows the structural model, which reveals the significance of the relationship between the constructs. Details such as path coefficients and their *t*-values for the hypotheses and the coefficients of determination ($R^2$) for the dependent constructs are indicated in Table 6.

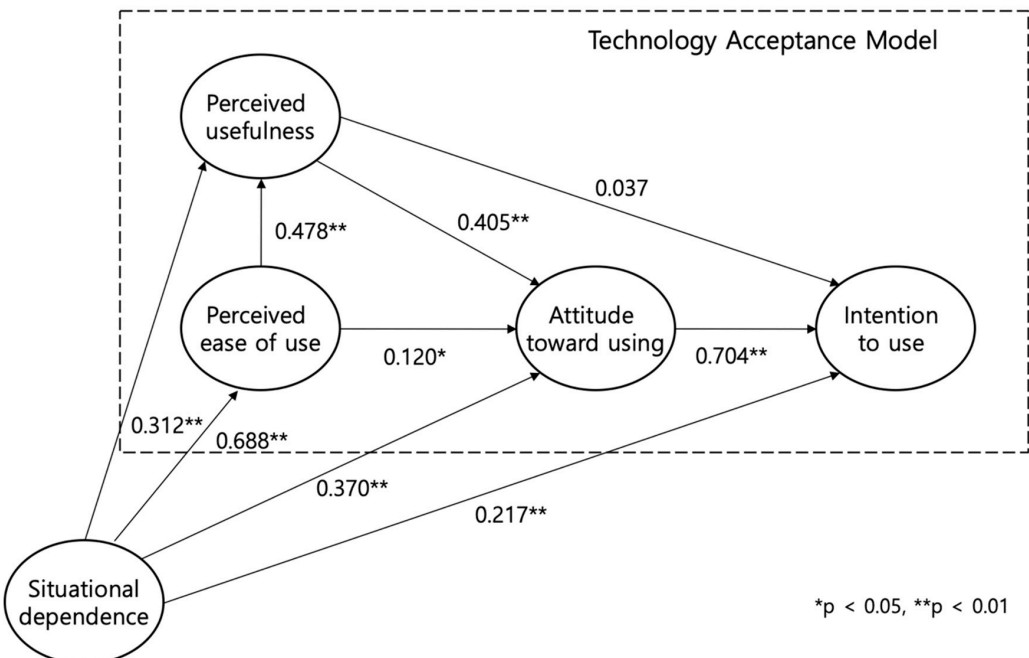

**Figure 2.** Path diagram for the research model.

**Table 6.** Hypothesis testing results.

| | Hypothesis | Sign | Path Coefficient | *t*-Value | *p*-Value |
|---|---|---|---|---|---|
| H1. | Situational Dependence → Perceived Usefulness | (+) | 0.312 | 5.216 | 0.000 |
| H2. | Situational Dependence → Perceived Ease of Use | (+) | 0.688 | 18.691 | 0.000 |
| H3. | Situational Dependence → Attitude toward Using | (+) | 0.370 | 5.101 | 0.000 |
| H4. | Situational Dependence → Intention to Use | (+) | 0.217 | 4.135 | 0.000 |
| H5. | Perceived Ease of Use → Perceived Usefulness | (+) | 0.478 | 7.547 | 0.000 |
| H6. | Perceived Ease of Use → Attitude toward Using | (+) | 0.120 | 1.746 | 0.041 |
| H7. | Perceived Usefulness → Attitude toward Using | (+) | 0.405 | 5.390 | 0.000 |
| H8. | Perceived Usefulness → Intention to Use | (+) | 0.037 | 0.523 | 0.301 |
| H9. | Attitude toward Using → Intention to Use | (+) | 0.704 | 11.744 | 0.000 |

Perceived Usefulness $R^2$: 0.530; Attitude toward Using $R^2$: 0.636; Intention to Use $R^2$: 0.811.

We performed the hypothesis testing based on the structure model. As indicated in Table 6, the results showed that situational dependence significantly affects perceived usefulness, perceived ease of use, attitude toward using, and intention to use; perceived ease of use significantly affects perceived usefulness; perceived usefulness significantly affects attitude toward using; and attitude toward using significantly affects intention to use, with $\alpha = 0.01$. Thus, H1, H2, H3, H4, H6, H7, and H9 were supported. H8, which indicates that perceived usefulness has a significant effect on intention to use, was rejected.

About 53% of the variance in perceived usefulness ($R^2 = 0.530$) was explained by situational dependence and perceived ease of use; 63.6% of the variance in attitude toward using ($R^2 = 0.636$) was explained by situational dependence, perceived ease of use, and perceived usefulness; and 81.1% of the variance in intention to use ($R^2 = 0.811$) was explained by situational dependence, perceived usefulness, and attitude toward using.

## 6. Discussion and Contributions

To emphasize the importance of the situation in individuals' acceptance of SST, this study developed the situational dependency construct and empirically analyzed the role of situational dependency in the use of SST. The results showed that situational dependence on SST positively affects perceived usefulness, attitude, and intention to use the technology. Thus, all of the hypotheses related to situational dependence were accepted.

These results suggest that situational factors, such as time pressure and complex usage environment, play an important role in the use of SSTs aside from the usefulness of the technologies in performing tasks. Although SSTs are voluntary technologies, they are mostly used in mandatory tasks, such as flight check-in and supermarket check-out. In this mandatory processing of tasks, people seem to consider that using SST is an effective alternative in situations such as queuing. In other words, situation dependence can be confirmed to be an important factor in the use of SSTs. Another interpretation of the findings is that the study was conducted in a specific area of the hospital, and it is possible that the subjects of the study were highly situationally dependent on the SST provided by the hospital because they were already familiar with it or felt a sense of belonging in regards to the hospital.

The hypothesis testing of this research model showed that perceived usefulness has no direct effect on intention to use. This result suggests that SSTs may not be that useful because they are used immediately and that obstacles such as technology anxiety [9] are in the way of SST use.

*6.1. Contributions and Implications*

This study provides the following meaningful theoretical contributions. Previous research on SST acceptance has focused primarily on the nature of the technology and on the characteristics of the individuals who use it. Research on the situational factors is limited. In this study, we developed the construct of situational dependence based on situational factors and verified the influence of situational dependence through the SST acceptance model. Therefore, the results of this study are able to serve as a theoretical basis for making research models in subsequent studies that analyze the influence of situational factors for accepting SST.

This study also has important practical implications for accepting the SST of individuals. The results show that, along with perceived ease of use and perceived usefulness, situational dependence influences attitude toward using and that situational dependence strongly affects intention to use. Although ease of use and usefulness of technology are considered to be important issues in traditional SST environments, the situation also plays an important role in accepting SSTs. Therefore, SST operators are encouraged to increase their customers' initial use of SSTs by increasing the situational dependence on SSTs.

*6.2. Limitations and Recommendations for Future Research*

While the results of this study make theoretical contributions to research and have important implications for practitioners, the study has the following research limitations. First, this study did not involve various factors that could influence the acceptance of SSTs by individuals, such as technology anxiety [9], enjoyment [20], and self-efficacy [10]. Future studies should involve these factors in their research models and empirically test the model to better understand the intention to use SSTs. Second, this study only examined the effects of situational dependence in the setting of hospital kiosks. To generalize the results of the research, future studies should examine the effect of situational dependence on SSTs in different areas, such as banks, hotels, and supermarkets. Lastly, further research should develop more valid measurement items for the situational dependence construct.

**Author Contributions:** Conceptualization, C.Y. and B.C.; methodology, C.Y. and B.C.; formal analysis, C.Y. and B.C.; investigation, B.C.; data curation, B.C.; writing—original draft preparation, C.Y.; writing—review and editing, C.Y. and B.C.; supervision, C.Y. All authors have read and agreed to the published version of the manuscript.

**Funding:** This research received no external funding.

**Conflicts of Interest:** The authors declare no conflict of interest.

**Appendix A**

**Perceived Usefulness: Likert scale ranging from strongly disagree to strongly agree**

a1. The hospital's self-service kiosk is effective in performing my job.
a2. Using the hospital's self-service kiosks improves my job performance.
a3. I consider the self-service kiosks of the hospital to be useful.

**Perceived Ease of Use: Likert scale ranging from strongly disagree to strongly agree**

a4.　My interaction with the hospital's self-service kiosks is clear and understandable

a5.　I think the hospital self-service kiosks are easy to use.

a6.　When using the hospital's self-service kiosks, not much effort is needed to interact with the machine.

**Situational Dependence: Likert scale ranging from strongly disagree to strongly agree**

a7.　I think that doing things without the help of self-service kiosks will take much more time when there are many people.

a8.　I sometimes think that using self-service kiosks is a good way to get my job done easily.

a9.　I usually feel that performing my job would not be easy without self-service kiosks.

**Attitude toward Using: Likert scale ranging from strongly disagree to strongly agree**

a10.　Using the hospital's self-service kiosks is a good idea.

a11.　It is good to use the hospital's self-service kiosks.

**Intention to Use: Likert scale ranging from strongly disagree to strongly agree**

a12.　I will use the hospital's self-service kiosks in the future.

a13.　I will enjoy using the hospital's self-service kiosks.

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
