# Peer review of "Role of Situational Dependence in the Use of Self-Service Technology"

_sustainability, doi:10.3390/su12114653_

Round 1

Reviewer 1 Report

I consider the topic is exciting. However, I recommend several changes to improve the final product.
1) A spell check is required. I found several typographical errors:
1.a) In section 2.2, on line 117, it seems that in the sentence "... it much time to process ..." a verb is missing. Sounds better "... it's a long time to process ..."
1.b) In section 3.1, in figure 1, it is necessary to change "H12" to "H2".
1.c) In section 5.1, on line 216, it is necessary to separate “3areover”.
1.d) In section 5.3, on line 251, it is necessary to separate “and81.1%”.
2) Comments to section 1 (Introduction):
I think the authors should mention that TAM is widely used in two versions: TAM with attitude and TAM without attitude; none of them seems to prevail over the other. To appreciate this, I recommend the article entitled "The role of attitudes in TAM: a theoretically unnecessary construction?" published in British Journal of Educational Technology, 2011, Vol 42, No 6, E160 - E162. Probably applying your data to these two models would add more value to your paper.
3) Comments to section 2.1 (Self-service technology):

3.a) Table 1 is incomplete. Many factors have not been included, such as: I) Cognitive factors (beliefs in entertainment, confort, control, and delivery speed). I recommend reading the article “Self-Service Technology Versus Traditional Service: Examining Cognitive Factors in The Purchase Of The Airline Ticket”, Journal of Travel & Tourism Marketing, 2013, 30, 497–508; II) Sensation-seeking. III) Innovativeness. IV) Self-consciousness.
3.b) I don't quite understand the difference between the method dimension and the task-processing dimension. I think it is convenient to explain a little more or unite these two dimensions in one.
4) Comments to section 3 (Research model and Hypotheses): If you have studied the relationships between "situational dependency" and the different TAM variables, I wonder why you have not also examined the relationship between "situational dependency" and "perceived ease of use."
5) Comments to sections 4 and 5 (Research methodology and Results):

5.1) I recommend moving the first paragraph from the Results section (lines 194 to 198) to the Research Methodology section. You might call it "Data Analysis Techniques".

5.2) I would like the authors mention that there are two approaches to modelling structural equations; they are covariance‐based SEM or CB‐SEM and variance‐based SEM or PLS‐SEM.

Author Response

We would first like to state that we appreciate your thoughtful comments. We have endeavored to follow your advice.

I consider the topic is exciting. However, I recommend several changes to improve the final product.

1) A spell check is required. I found several typographical errors:

  1. a) In section 2.2, on line 117, it seems that in the sentence "... it much time to process ..." a verb is missing. Sounds better "... it's a long time to process ..."
  2. b) In section 3.1, in figure 1, it is necessary to change "H12" to "H2".
  3. c) In section 5.1, on line 216, it is necessary to separate “3areover”.
  4. d) In section 5.3, on line 251, it is necessary to separate “and81.1%”..

We appreciate the favorable comments of our paper and a thoughtful point in the errors. We corrected the mistake. Also, we have performed several reviews on the manuscript and improved the grammar and syntax of the content of the manuscript.

2) Comments to section 1 (Introduction):

I think the authors should mention that TAM is widely used in two versions: TAM with attitude and TAM without attitude; none of them seems to prevail over the other. To appreciate this, I recommend the article entitled "The role of attitudes in TAM: a theoretically unnecessary construction?" published in British Journal of Educational Technology, 2011, Vol 42, No 6, E160 - E162. Probably applying your data to these two models would add more value to your paper.

Thank you for your accurate comments and for providing relevant references (López-Bonilla & López-Bonilla 2011). As you mentioned, today, the TAM theory uses a mixture of the version except for the attitude variable and the original version including the attitude variable. Since Davis, Bagozi & Warshaw (1989) pointed out unnecessary for attitude variables in TAM in their early TAM comparative study, recent studies using the TAM mainly use the version excluding the attitude variable, However, in the case of SSTs, many studies (Dabholkar 1996, Dabholkar & Bagozzi 2002, Curran et al. 2003, Curran & Meuter 2005, Liu et al. 2012) used the attitude variable because of the technical characteristics of SSTs and the high explanatory power of attitude toward intention to use. Therefore, we also used the attitude variable.

We included the following paragraph in the Research model section.

Today, the TAM theory uses a mixture of the version except for the attitude variable and the original version including the attitude variable (López-Bonilla & López-Bonilla 2011). Although recent studies using the TAM mainly use the version excluding the attitude variable, in the case of SSTs, many studies (Dabholkar 1996, Dabholkar & Bagozzi 2002, Curran et al. 2003, Curran & Meuter 2005, Liu et al. 2012) used the attitude variable because of the technical characteristics of SSTs and the high explanatory power of attitude toward intention to use. Therefore, this study also uses the attitude variable.

3) Comments to section 2.1 (Self-service technology):

  1. a) Table 1 is incomplete. Many factors have not been included, such as: I) Cognitive factors (beliefs in entertainment, comfort, control, and delivery speed). I recommend reading the article “Self-Service Technology Versus Traditional Service: Examining Cognitive Factors in The Purchase Of The Airline Ticket”, Journal of Travel & Tourism Marketing, 2013, 30, 497–508; II) Sensation-seeking. III) Innovativeness. IV) Self-consciousness.

Thank you for your meticulous review. We added the cognitive factors that are entertainment (as enjoyment), comfort, control, and delivery speed (López-Bonilla & López-Bonilla 2013), also included sensation-seeking, innovativeness (Lin & Chang 2011, López-Bonilla & López-Bonilla 2012, Vakulenko et al. 2019), and self-consciousness (Dabholkar & Bagozzi 2002) into Table 1.

  1. b) I don't quite understand the difference between the method dimension and the task-processing dimension. I think it is convenient to explain a little more or unite these two dimensions in one.

We appreciate your constructive comments. We have redefined below so that readers can more clearly distinguish between the method dimension and the task-processing dimension.

Method dimension: the degree of dependence on SSTs because of the perception that the methods and procedures for performing a task are complicated and difficult without using SSTs in the present situation

Task-processing dimension: the degree of dependence on SSTs because of the perception that it is impossible to achieve a task without using SSTs in the present situation

4) Comments to section 3 (Research model and Hypotheses): If you have studied the relationships between "situational dependency" and the different TAM variables, I wonder why you have not also examined the relationship between "situational dependency" and "perceived ease of use."

Following your advice, we added a hypothesis on the relationship between situational dependence and perceived ease and examined the hypothesis. The added hypothesis is as follows;

H2: Situational dependence positively affects perceived ease of use.

As expected, the hypothesis was accepted. We described the details of the analysis results in the Hypothesis Testing section.

5) Comments to sections 4 and 5 (Research methodology and Results):

  1. a) I recommend moving the first paragraph from the Results section (lines 194 to 198) to the Research Methodology section. You might call it "Data Analysis Techniques".

According to your recommendations, we moved the paragraph from the Results section to the Research Methodology section and created 4.3. Data Analysis Techniques subsection

  1. b) I would like the authors mention that there are two approaches to modelling structural equations; they are covariance‐based SEM or CB‐SEM and variance‐based SEM or PLS‐

We have added the following paragraph to 4.3. Data Analysis Technique.

This study uses structural equation modeling (SEM) for analysis of the research model. SEM, a statistical analysis method mainly used for causality analysis today, is divided into covariance-based structural equation modeling (CB-SEM) and partial least squares structural equation modeling (PLS-SEM). Unlike CB-SEM, which is mainly used for confirmatory research, PLS-SEM is advantageous for conducting exploratory studies and analyzing complex models because the requirements for sample size or residual distribution are relatively less stringent (Hair et al. 2014). Therefore, PLS-SEM has been actively used in various fields. Since this study includes exploratory nature and the PLS-SEM are more flexible than the CB-SEM in the study, this study employs the PLS-SEM approach.

Reference

  1. Curran, J. M., & Meuter, M. L. (2005). Self-service technology adoption: comparing three technologies. Journal of services marketing, 19(2), 103-113.
  2. Curran, J. M., Meuter, M. L., & Surprenant, C. F. (2003). Intentions to use self-service technologies: a confluence of multiple attitudes. Journal of Service Research, 5(3), 209-224.
  3. Dabholkar, P. A. (1996). Consumer evaluations of new technology-based self-service options: an investigation of alternative models of service quality. International journal of research in marketing, 13(1), 29-51.
  4. Dabholkar, P. A., & Bagozzi, R. P. (2002). An attitudinal model of technology-based self-service: moderating effects of consumer traits and situational factors. Journal of the academy of marketing science, 30(3), 184-201.
  5. Davis, F. D., Bagozzi, R. P., & Warshaw, P. R. (1989). User acceptance of computer technology: a comparison of two theoretical models. Management science, 35(8), 982-1003.
  6. Hair, J. F., Sarstedt, M., Hopkins, L., & G. Kuppelwieser, V. (2014). Partial least squares structural equation modeling (PLS-SEM) An emerging tool in business research. European Business Review, 26(2), 106-121.
  7. Lin, J. S. C., & Chang, H. C. (2011). The role of technology readiness in self-service technology acceptance. Managing Service Quality, 21(4), 424–444.
  8. Liu, S. F., Huang, L. S., & Chiou, Y. H. (2012). An integrated attitude model of self-service technologies: evidence from online stock trading systems brokers. The Service Industries Journal, 32(11), 1823-1835.
  9. López-Bonilla, L. M., & López-Bonilla, J. M. (2011). The role of attitudes in the TAM: A theoretically unnecessary construct?British Journal of Educational Technology, 42(6), E160-E162.
  10. López-Bonilla, J. M., & López-Bonilla, L. M. (2012). Sensation-Seeking Profiles and Personal Innovativeness in Information Technology. Social Science Computer Review, 30(4), 434–447.
  11. López-Bonilla, J. M., & López-Bonilla, L. M. (2013). Self-service technology versus traditional service: Examining cognitive factors in the purchase of the airline ticket. Journal of Travel & Tourism Marketing, 30(5), 497-508.
  12. Vakulenko, Y., Oghazi, P., & Hellström, D. (2019). Innovative framework for self-service kiosks: Integrating customer value knowledge. Journal of Innovation and Knowledge, 4(4), 262–268.

Reviewer 2 Report

The proposed research is well suited to today's environment. However, the findings do not entirely match the ambitions announced by the abstract.

The design is evident within the announced theoretical framework. However, the social cognition field is much more nuanced when discussing situational influence on the behaviour. Although announced as a confirmatory study, it seems to be more an exploratory one. The data collection context is confined to a particular situation, a specific social context having a significant number of respondents being employs within the same organization. The analyzed behaviour might be, probably, influenced by the familiarity and the perception of belongings. Taking into account these factors during the analyze and the discussion section would make the research much more clear.

Author Response

We would first like to state that we appreciate your thoughtful comments. We have endeavored to follow your advice.

The proposed research is well suited to today's environment. However, the findings do not entirely match the ambitions announced by the abstract.

We agree with your comments. This study is an initial study that explores the role of situational dependence in information technology use. Therefore, the level of the findings is not as high as expected. Please kindly understand this situation.

The design is evident within the announced theoretical framework. However, the social cognition field is much more nuanced when discussing situational influence on the behaviour. Although announced as a confirmatory study, it seems to be more an exploratory one. The data collection context is confined to a particular situation, a specific social context having a significant number of respondents being employs within the same organization. The analyzed behaviour might be, probably, influenced by the familiarity and the perception of belongings. Taking into account these factors during the analyze and the discussion section would make the research much more clear...

Thank you for your constructive opinion. In accordance with your opinion, we have included the following sentences in the Discussion and Contributions section.

Another interpretation of the findings is that the study was conducted in a specific area of the hospital, and it is possible that the subject of the study was highly situationally dependent on the SST provided by the hospital because they were already familiar with or felt a sense of belonging to the hospital.

Reviewer 3 Report

There are some minor corrections of this paper including typos(e.g. page 10 line 216 Table 3areover ) and comma in citations (e.g. page 3 line 76 Meuter et al."," 2003). 

The TAM is one most adopted model in technology adoption research. The research only evaluate  the  path between  SD and Usefulness, the relationship between SD and EOU is also recommend to evaluate.   

Author Response

We would first like to state that we appreciate your thoughtful comments. We have endeavored to follow your advice.

There are some minor corrections of this paper including typos(e.g. page 10 line 216 Table 3areover ) and comma in citations (e.g. page 3 line 76 Meuter et al."," 2003).

We appreciate the favorable comments of our paper and a thoughtful point in the errors. We corrected the mistake. Also, we have performed several reviews on the manuscript and improved the grammar and syntax of the content of the manuscript.

The TAM is one most adopted model in technology adoption research. The research only evaluate the path between SD and Usefulness, the relationship between SD and EOU is also recommend to evaluate.

Following your advice, we added a hypothesis on the relationship between situational dependence and perceived ease of use, and examined the hypothesis. The added hypothesis is as follows:

H2: Situational dependence positively affects perceived ease of use.

As expected, the hypothesis was accepted. We described the details of the analysis results in the Hypothesis Testing section.

Round 2

Reviewer 1 Report

I think the authors have responded to all recommendations. Thank you for the changes in your article.

Author Response

I think the authors have responded to all recommendations. Thank you for the changes in your article.

Thank you very much for your efforts in improving the quality of our paper.

Reviewer 2 Report

Thank you for taking into account the comments.

There is a possible inappropriate self-citations - please check the "Cheolho Yoon, Dongsup Lim, Changhee Park, Factors affecting adoption of smart farms: The case of Korea, Computers in Human Behavior, Volume 108, 2020, 106309, ISSN 0747-5632, https://doi.org/10.1016/j.chb.2020.106309".

It might be the case of a mistake, but anyway, the article is not mentioned in the Reference part and, due to the similarity detected, the citation shall be revised. 

Author Response

There is a possible inappropriate self-citations - please check the "Cheolho Yoon, Dongsup Lim, Changhee Park, Factors affecting adoption of smart farms: The case of Korea, Computers in Human Behavior, Volume 108, 2020, 106309, ISSN 0747-5632, https://doi.org/10.1016/j.chb.2020.106309".

It might be the case of a mistake, but anyway, the article is not mentioned in the Reference part and, due to the similarity detected, the citation shall be revised.

Thank you for your careful consideration.

In this paper, we did not refer to the article. We knew that some of the sentences (especially the methodology section) of the article were found in our paper during a plagiarism check. As the journal office pointed out, we have rewritten the sentences found in the plagiarism check. As a result, there are no reused sentences in this study ever, also no self-citations to the article.